MINIREVIEW

# What Lies Beneath? Taking the Plunge into the Murky Waters of Phage Biology

Mirjam Zünd,ᵃ Sage J. B. Dunham,ᵃ Jason A. Rothman,ᵃ Katrine L. Whitesonᵃ

ᵃDepartment of Molecular Biology and Biochemistry, University of California, Irvine, California, USA

**ABSTRACT** The sequence revolution revealed that bacteria-infecting viruses, known as phages, are Earth's most abundant biological entities. Phages have far-reaching impacts on the form and function of microbial communities and play a fundamental role in ecological processes. However, even well into the sequencing revolution, we have only just begun to explore the murky waters around the phage biology iceberg. Many viral reads cannot be assigned to a culturable isolate, and reference databases are biased toward more easily collectible samples, which likely distorts our conclusions. This minireview points out alternatives to mapping reads to reference databases and highlights innovative bioinformatic and experimental approaches that can help us overcome some of the challenges in phage research and better decipher the impact of phages on microbial communities. Moving beyond the identification of novel phages, we highlight phage metabolomics as an important influencer of bacterial host cell physiology and hope to inspire the reader to consider the effects of phages on host metabolism and ecosystems at large. We encourage researchers to report unassigned/unknown sequencing reads and contigs and to continue developing alternative methods to investigate phages within sequence data.

**KEYWORDS** genomic dark matter, metabolomics, metagenomics, phage

## PHAGES, THE MOST ABUNDANT BIOLOGICAL ENTITY, SHAPE BACTERIAL COMMUNITIES

Despite their omission from the phylogenetic tree of life, viruses that infect bacteria (bacteriophages, or phages) have a profound influence on the shape of the tree itself. Through predation and gene exchange, phages have steered the evolution of bacteria, perhaps as much or more than any other selective factor (1). Beyond evolution, phages drive ecosystem dynamics in the present, partially dictating bacterial gene expression and metabolism and whether or not a bacterial species can survive to pass its genes on to its progeny (2–4). Found in nearly every environment, phages are the most abundant entities in our biosphere, with an estimated population of $\sim 10^{31}$ particles (5).

Far from an abstract scientific curiosity, this viral abundance presents an amazing opportunity for understanding and influencing the ecosystems around, on, and inside us. Viruses lyse an estimated 20 to 30% of cells in the ocean and thus enormously affect the marine food web and carbon cycle (3, 6). Phages offer a unique opportunity to engineer both host-associated and environmental microbiomes and will have a profound impact on how we interact with the microbial world in the 21st century. For example, in response to the unrelenting rise in antibiotic-resistant bacterial infections, the 100-year-old strategy of phage therapy has reemerged as a promising alternative to antibiotics (7, 8). In another example, when used as part of fecal filtrate transplantation, phages modulate bacterial colonization in the intestine and contribute to restoring a more beneficial microbial community (9). Owing to their capacity for rapid evolution and propensity to transfer essential genes and functions to their microbial hosts during times of stress, the genetic information carried by phages and their hosts are akin to canaries in a coal mine—telegraphing the presence of environmental stressors (10, 11).

Address correspondence to Katrine L. Whiteson, katrine@uci.edu.

Conflict of Interest Disclosures for the Authors: Mirjam Zünd reports grants from University of California Office of the President during the conduct of the study. Sage J. B. Dunham reports a training grant from the National Institute of Aging during the conduct of the study. Jason A. Rothman reports grants from University of California Office of the President and a Hewitt Foundation Postdoctoral Fellowship during the conduct of the study. Katrine L. Whiteson reports grants from University of California Office of the President, a training grant from the National Institute of Aging, and a Hewitt Foundation Postdoctoral Fellowship during the conduct of the study.

Conflict of Interest Disclosures for the Editor: Jack A. Gilbert is a Scientific Advisory Board Member for DayTwo.

*This minireview went through the journal's normal peer review process. DayTwo sponsored the minireview and its associated video but had no editorial input on the content.*

Generating a comprehensive picture of phages and their biological capacities using traditional isolation approaches is challenging, since both the phage and its microbial host must be culturable under laboratory conditions (12). As a result, our knowledge about phages is necessarily biased toward the small fraction of easily accessible and culturable viruses (13). Even now, with all of the advantages offered by high-throughput sequencing, up to 90% of viral metagenomic sequence reads are unassignable, meaning that they lack matches to nucleotide or protein databases, inspiring terms such as "viral dark matter" (5, 6). Furthermore, the nucleotide sequences (which are generated almost entirely from isolated phages) in reference databases are substantially dissimilar to most phages discovered via cultivation-independent sequence-based approaches (14–16). Remarkably, 75% of the available isolated phage genomes are associated with only 30 bacterial genera (13), and concerted sequencing efforts continue to yield many novel phages, highlighting that phage discovery has yet to reach saturation (17, 18). Performing analysis on only phages that are closely related to isolated phages is unlikely to accurately represent the true composition, distorting the overall interpretation of the ecosystem under study. Supporting conclusions with a larger fraction of the available data will provide a more accurate approximation of the true biological system. Therefore, besides the effort to include viruses in microbiome studies, we must go further and apply strategies that go beyond read alignment to cultivable phages to account for those unassignable sequences which do not match any isolated sequences. Reads not mapping to isolated phages should be routinely included in analyses of shotgun data to drive discoveries.

In this minireview, we provide a brief overview of phage characterization efforts as they stand today. We discuss sequencing efforts in detail, particularly those that move beyond standard reference databases. We also discuss experimental approaches for expanding our understanding of phage biology and touch on the exciting potential to use metabolomics to characterize phage infections, which we expect to yield powerful discoveries in the years to come (Fig. 1). As this is a minireview, there are many topics that we could not cover in sufficient depth. Please see reference 19 for an overview of the assembly and annotation process for phages and reference 20 for an insight into the use of metatranscriptomics to study the diversity of RNA viruses.

## BULK AND VIROME SEQUENCES TARGET DIFFERENT POPULATIONS WITHIN THE UNCULTIVATABLE PHAGE FRACTION

Starting in 1976 with the RNA phage MS2, phages were the first organisms to have their genomes completely sequenced (21). As with other microorganisms, phage discovery has been greatly aided by recent advancements in sequencing technology. Today, sequencing represents the principal approach for discovering novel phages, and since phages lack universal phylogenetic marker genes (22), bulk (metagenome) and enriched virome sequencing are the methods primarily used for phage detection. Generally, during bulk sequencing, bacteria and phages are not separated, which results in the sequencing of genomes from both phages found within the host cell (e.g., prophages and replicating phages) and virions. However, because phage genomes are much shorter than bacterial genomes, phages represent only a minor fraction of all the genetic material in a given sample, so detection is often restricted to highly abundant particles. Virome sequencing overcomes these obstacles by physically enriching virions before sequencing through a combination of filtration, precipitation, and nuclease treatment. However, viromes largely miss prophages and replicating phages trapped in the host cell. Most enrichment methods are also limited in the phages they can target due to various biochemical properties (e.g., RNA versus DNA viruses, different capsid proteins or membrane envelopes) and introduce compositional biases due to the need for DNA amplification. For these reasons, virome sequencing provides a qualitative but not quantitative survey of the phage community (23–25). Furthermore, before starting an experiment, the researcher must decide which phages (i.e., host-associated phages versus virions and RNA versus DNA) are of interest. It is often necessary to perform multiple

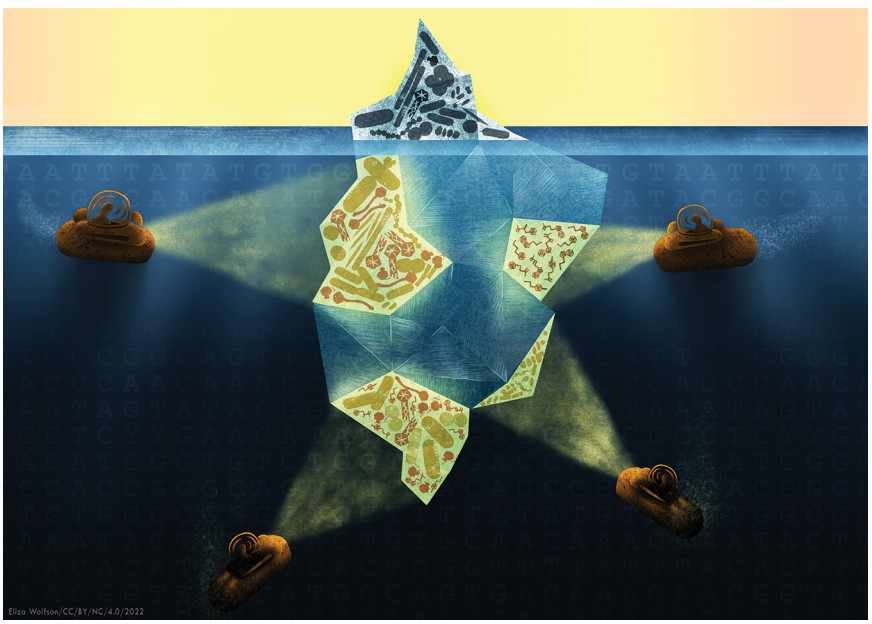

**FIG 1** If, by way of analogy, we consider the diversity of phages in our biosphere as a giant iceberg, then our current knowledge is represented by only the small visible portion of the iceberg. The discrepancy between the estimated phage abundance and our current knowledge indicates that most information about phage diversity and its effects on their hosts and ecosystems remains obscured underneath the surface. We chose to highlight four topics (symbolized by the submarines) that will help us to explore this murky world of phage biology. (i) Virome and bulk (metagenomic) sequencing methods are the main driving forces behind modern phage discovery efforts, helping us to uncover new phages and understand their impact on the microbial world. Applying *de novo* assembly and viral feature identification helps us to reduce the pile of undiscovered phage genomes further and expand the content of so-called alternative databases. (ii) Novel bioinformatic tools allow us to investigate the activity state of prophages and quantify their induction, providing insights into the impact of phages on microbial populations. These tools can provide a starting point for designing studies to disentangle the effect of phages on the ecosystem. (iii) By combining experimental and sequence-based approaches, we can tackle questions that are challenging or impossible with either strategy individually. One of the most successful examples is phage-host range analysis, where these methods shed light on the phage-host interaction network. (iv) Although the topic affects almost every arena where phages are involved, the impact of phage infection on hosts' metabolic state is woefully understudied. Except in a few cases, little is known about how phages affect host metabolism, thus providing considerable potential for discoveries that will influence everything from medicine to climate change. (Image courtesy of Eliza Wolfson, @eliza_coli.)

workflows, for example, two different library preparations for RNA and DNA phages and virome and bulk sequencing to capture both host-associated and free virions. Although both bulk and virome sequencing methods detect unculturable phages in the microbiota, they target different populations with a minor overlap (15, 26).

## STRATEGIES FOR STUDYING PHAGE GENOMICS OUTSIDE OF TRADITIONAL REFERENCE DATABASES

Sequence reads can be identified as viral either individually or after assembly into contigs that represent complete or partial virus genomes. Pairwise alignment of reads and contigs to reference databases allows for the detection of viral sequences. Alternatively, viral protein sequences can be aligned against viral protein databases to find homologs and assign taxonomy. However, homology searches are limited by highly divergent protein sequences (less than 30% identity to the query), which is often the case for viruses because of their higher substitution and mutation rates compared to other organisms (27). Profiling methods like hidden Markov models provide the opportunity to classify sequences as viral or nonviral and even compare them to those of distantly related phages (28, 29). Such profiling models consider information across a family of evolutionarily related sequences to incorporate position-specific information about variation across family members into the alignments and thus are well suited to identifying divergent viral sequences.

Even though hidden Markov models are more likely to identify novel phages than reads or contig mapping, identifying sequence reads originating from viruses absent from reference databases can be difficult. However, there are several promising approaches for detecting and studying novel viral sequences.

**De novo assembly and screening of virus features to identify novel viruses.** *De novo* assembly of viral contigs is the first step to identifying sequences of viral origin without mapping to reference sequences (30, 31). Assembling data sets from different samples, i.e., cross-assembly, is particularly powerful. For example, this approach was used to discover crAssphage, which, despite being a highly abundant intestinal phage family, remained hidden because its proteins did not match those in reference databases (32). Contigs can be screened for viral characteristics to verify that they originate from viruses rather than bacterial contamination or assembly artifacts. Such characteristics include signs of circularity (an indicator of a complete genome with the caveat of excluding linear phages), encoded viral protein families, viral nucleotide signatures (e.g., small genes, high coding density, few strand switches), or encoded lineage-specific marker genes. In one good example of this approach, Benler et al. filtered viral genomes using the abovementioned features combined with phylogenetic analysis and gene-sharing networks to detect previously undescribed phage lineages, greatly increasing the characterized diversity of the human gut phage community (33). Differentiation of bacterial and viral contigs is a crucial step in exploring phages, and several tools have been developed using gene content or motifs (i.e., k-mer frequency) that simplify viral identification (34–40). To some extent, phage identification tools rely on features from previously isolated phages; nevertheless, they have been instrumental in the identification of thousands of novel phages. In turn, these tools have enabled the construction of several uncultivated phage databases (referred to as "alternative databases"; e.g., HuVirDB, GVD, IMG/VR v3, MGV) (18, 26, 41, 42). The collection of uncultivated phage genomes greatly outnumbers the genomes from isolated phages (18, 26, 42, 43), with alternative databases containing up to 12 times more high-quality genomes than reference databases or even 100 times more when including fragmented genomes of any quality (13, 18). Thus, alternative databases offer a rich resource for the target genome when using read mapping to identify or filter phages from background bacterial contamination (44–46).

**Reducing the unassembled fraction of bulk metagenome reads by the targeted use of virome data.** *De novo* assembly of bulk data does not guarantee the inclusion of all reads to bacterial and viral genomes. Reads from low-abundance phages will likely not have adequate coverage for assembly and will be missed during phage identification if the sequencing depth is not drastically increased. Considering the abundance of phages in our environment, many unassembled reads likely originate from low-abundance virions (47). One low-cost approach for identifying phages within the unassembled bulk reads is to pool samples before virion enrichment and sequencing, which enables improved coverage for low-abundance viruses. Mapping bulk data to the viral contigs generated from the pooled virome sample might reveal low-abundance viruses in individual bulk data, allowing us to identify otherwise missed phages.

## DETECTING INDUCIBLE PROPHAGES TO INVESTIGATE THEIR IMPACT ON THE MICROBIOME

At least half of the sequenced human intestinal bacteria contain temperate phages integrated into their genomes (i.e., prophages) (48, 49). One common strategy to identify induced prophages in virome data is to search the viral contigs for prophage hallmark genes such as integrases and excisionases. Although indisputably valuable, this process risks distorting temperate phage classification, since obligate lytic phages (such as T7) also encode integrases and not all prophages rely on an integrase for their insertion (50–53). For example, an analysis by Silveira et al. found that only two-thirds of the predicted prophages in their data encoded commonly used integration marker genes, while one-third did not (54).

The association of prophages with their hosts makes bulk sequencing the preferred method for prophage identification. Prediction tools that identify prophages in host genomes have greatly advanced our understanding of prophage prevalence. However, most classical phage identification tools cannot differentiate between intact or cryptic prophages. Therefore, several bioinformatic approaches were recently developed to identify inducible prophages by mapping bulk reads to reference or assembled host genomes. One approach identifies active prophages by screening for regions with elevated read coverage relative to the host genome to indicate replicating phages (55, 56). Another approach uses unusual read alignment patterns resulting from genome circularization and concatemer formation during prophage replication (57). The alignment pattern enables the precise pinpointing of the prophage's location within its host genome. Both techniques enable quantifying phage activity by analyzing the ratio of the read coverage between phage and host bacterium (55). These tools have great potential to complement prophage prediction by locating prophage boundaries, differentiating between active and inactive prophages, and using bulk sequence data to study the effect of environment on induction dynamics and the impact of induced prophages on microbial community composition.

## TARGETED EXPERIMENTAL APPROACHES TRANSFORMING OUR KNOWLEDGE OF PHAGE-HOST SPECIFICITY

Determining a sequence read or contig to be of viral origin is of limited value for understanding community dynamics without information regarding host and activity. Linking uncultivated phages to their host and determining their host range has been a longstanding challenge in the field, and several *in silico* tools have been developed for this purpose (reviewed in references 58 and 59). An example on the nucleotide level is to search for host-encoded CRISPR spacers; matching to viral contigs allows the identification of phage-host pairs in large metagenomic data sets (42, 60). Another strategy is to predict phage-host pairs based on annotated receptor-binding proteins. Boeckaerts et al. developed a machine learning tool—based on the progress toward predicting receptor-binding proteins—to establish phage-host pairs for pathogenic organisms (61, 62). The recall and precision of such tools are limited by the completeness of the underlying databases, resulting in ambiguous performance. However, integrating several bioinformatic methods for detecting phage-host pairs indicates a promising improvement (63). Besides *in silico* tools, experimental approaches that do not rely on detecting host lysis (i.e., via the formation of phage plaques) have had some success. For example, combining primary virus enrichment, either by cultivation or by phage adsorption to bacterial cell debris, with virome sequencing has identified novel phages and their hosts (64, 65). Another extremely promising method for identifying phage-host pairs is proximity ligation sequencing, where genetic material that is in physical contact can be linked during library preparation (66, 67). This incorporation of spatial information with *de novo* assembly of bacterial and viral genomes has unique potential for assembling virus genomes and establishing host links. Using information from Hi-C linkage enables the grouping of viral contigs associated with the same host cell. Performing metagenomic binning within these groups enables reconstruction of genomes with higher overall quality and completeness than conventional workflows (66, 67). Moreover, it captures the interaction of phages with their respective host and thus is a powerful tool for discovering phage-microbe pairs and evaluating the phage host range within a community. Proximity ligation sequencing has the potential to transform what we know about phage host specificity. Currently, there are rarely more than hundreds of bacteria included in a typical phage host range study; however, with proximity ligation sequencing, it is possible to assess the phage host range within an entire complex and even uncultivatable sample (67). For example, Marbouty et al. showed that 17 members of the crAss-like phage infect different strains of *Bacteroidetes* (68).

## UNDERSTANDING THE METABOLIC CONSEQUENCES OF A PHAGE INFECTION

Reliant upon the biological mechanics of their hosts, phages cannot autonomously perform many processes considered fundamental to life, including metabolism. Phages must therefore rely entirely on their hosts for the production of all necessary biomolecules. Although much can be learned through the functional characterization of a phage genome, a more comprehensive understanding of the metabolic consequences of a phage infection can be achieved through metabolomics experiments conducted over the course of a phage infection. In some instances, novel phages may even be revealed through an examination of "phage-induced" metabolic phenotypes that arise in uncharacterized multicomponent systems. We further argue that just as the identification of a novel phage is incomplete without a specific link to a host, the characterization of a phage is incomplete without understanding its impact on host metabolism. We therefore broadly define phage metabolomics as "the study of phage-mediated metabolic changes in bacteria," a definition first espoused by others (e.g., De Smet [69]).

Although phage metabolomics may appear to be relatively new, it has been studied for decades, perhaps most notably arising with the characterization of the famous phage T4, which infects *E. coli* (70). During the initial stages of infection, phage T4 comprehensively remodels the host metabolism, forcing the cells to produce the large assortment of biomolecules necessary for the maximal production of virions. This remodeling process even forces degradation of the host DNA and the halting of cytosine production in favor of hydroxymethylcytosine, which T4 uses in its DNA (70). In another profound example, it is estimated that 20% to 40% of all bacteria at the surface of the ocean are infected at any given time, and these infected cells (termed "virocells") can exhibit wildly different metabolic states from their uninfected counterparts. In many cases, the metabolic processes of the virocells are fundamentally reprogrammed to satisfy the fitness needs of the phage, having a profound impact on ocean ecosystems (4).

Although most phage-mediated changes in bacterial metabolism occur via host genes, many metabolic functions are encoded into the genomes of the phages themselves. Phage-encoded metabolic genes, known as auxiliary metabolic genes (AMGs) (71), are thought to primarily assist in the phage replication process, but in some instances, they can have a profound influence on the larger ecosystem. For example, phages that prey on sulfur-metabolizing microbes contain numerous AMGs for the oxidation of sulfur and thiosulfate, contributing to biogeochemical cycling on a global scale (72). Bioinformatic approaches to identify AMGs in sequence data have been developed, providing insight into the metabolic potential of assembled phages (35). *In silico*, information about metabolic function might be used in the future to predict the success or failure of phage infection.

As phage therapy continues to gain traction as an alternative treatment for antibiotic-resistant bacteria, phage metabolomics will play an increasingly important role. Perhaps at the most basic level, metabolomics can help us to understand the biology underlying phage life cycles. For example, Anne Chevallereau and colleagues recently used metabolomics and transcriptomics to show that phage PAK_P3 successfully infects *Pseudomonas aeruginosa* by interfering with pyrimidine metabolism and forcing the generation of large quantities of the building blocks necessary for viral replication (73). Metabolites produced as a result of phage predation may act as adjuvants for phage and antibiotic therapy and could even be a source for entirely new antibiotics.

It is also important to understand the phage metabolic landscape from the perspective of safety. Prophages, in particular, are known to encode virulence factors and exotoxins, which they transcribe when exposed to an environmental stressor (e.g., antibiotic administration), presumably protecting the host bacteria (10). Although studies in this area are extremely limited, one tragic example is the colistin-induced release of Shiga toxin by *Escherichia coli*, which led to a fatal pulmonary exacerbation (74). Metabolomics revealed the 3-fold upregulation of the host's membrane receptor for Shiga toxin (74),

which is comprised of the small molecule globotriaosylceramide, in the days immediately preceding death. Therapeutic administration of lytic phages may also directly induce transcription of endogenous prophage genes and cause the production of virulence factors and exotoxins that harm a patient. Therefore, the safety screening for phage therapy should include a metabolic analysis of the phage-host combination.

## CONCLUSION

If properly harnessed, phages have the potential to manipulate microbial populations and help us address some of the 21st century's most pressing challenges in health care, environmental science, and industrial production. As the most plentiful infectious agent of the biosphere, the impact that phages have on living systems cannot be overstated. Thanks to incredible improvements in sequencing technology, the capacity of researchers to identify phages and understand their biology has greatly increased in recent years, but the tip of the iceberg is only beginning to emerge from the depths of the unknown (Fig. 1). Through database-dependent and database-independent approaches, we are uncovering the complicated interactions between phages and their hosts and shedding light on the "viral dark matter" that has long existed in sequencing experiments. Exploring this unknown viral ocean is a community effort, relying on support from everyone. Novel phages resolved from metagenomics should be actively deposited into databases to make them accessible. Currently, however, most of the alternative databases are not maintained, presenting only a snapshot of uncultivated viruses at the time of their compilation. A few databases, such as IMR/VR and the VIRION database (containing vertebrate viruses), are the exception and are updated to include novel uncultivated viruses (18, 75). Thus, to drive the field forward, maintaining common resources of uncultivated viruses is needed and will be an essential pillar of future progress. Moreover, reporting unassigned/unknown reads and contigs is vital to furthering our field. We are at an exciting time in phage biology and hope to inspire researchers to develop novel computational tools and methods that address these fascinating and complex members of biological communities. We truly are at the tip of the iceberg and invite everyone to dive into the viral unknown.

## ACKNOWLEDGMENTS

This work was supported by a grant from the University of California Office of the President Research Grants Program Office (award R00RG2814) to K.L.W., J.A.R., and M.Z. and by a Hewitt Foundation for Biomedical Research Postdoctoral Fellowship to J.A.R.; S.J.B.D. is supported by the NIA Aging and Alzheimer's Disease Training Grant (T32 AG00096-38).

We thank Heather Maughan for insightful feedback on the manuscript and Forest Rohwer, Anca Segall, and Rob Edwards for setting us on this path years ago.

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
