## [Reviewer comments · mSystems]

What lies beneath? Taking the plunge into the murky waters of phage biology

Mirjam Zünd, Sage Dunham, Jason Rothman, and Katrine Whiteson

Corresponding Author(s): Katrine Whiteson, University of California, Irvine

Review Timeline:

Submission Date:	August 24, 2022
Editorial Decision:	September 8, 2022
Revision Received:	November 1, 2022
Accepted:	November 2, 2022

Editor: Jack Gilbert

Reviewer(s): Disclosure of reviewer identity is with reference to reviewer comments included in decision letter(s). The following individuals involved in review of your submission have agreed to reveal their identity: Scott A Jackson (Reviewer #2)

Transaction Report:

DOI: <https://doi.org/10.1128/msystems.00807-22>

Prof. Katrine Whiteson
University of California, Irvine
Department of Molecular Biology and Biochemistry
School of Biological Sciences
3236 McGaugh Hall
Irvine, CA 92697-3900

Re: mSystems00807-22 (What lies beneath? Taking the plunge into the murky waters of phage biology)

Hey Katrine,

Reviewer comments are found at the end of this letter.

Your minireview is likely to be accepted once the indicated changes are made.

Author Bios: If you would like a brief biographical sketch of each author (limit, 150 words) to be published at the end of your article, please submit text and photos with your modified manuscript. For complete guidelines on submission requirements, please see the journal Submission and Review Process requirements at <https://journals.asm.org/journal/mSystems/submission-review-process>. **Submissions of a paper that does not conform to mSystems guidelines will delay acceptance of your manuscript.**

Figures [**Editor: insert figure numbers here**] in your manuscript are good candidates for graphical enhancement. We now offer our authors the services of ASM's contracted artist, Patrick Lane of ScEYence Studios. This art enhancement service is free of charge to authors of minireviews and full-length reviews, and turnaround time is fast. Please contact Patrick on receiving this letter. Complete contact information for Patrick and further instructions are posted at <https://journals.asm.org/pb-assets/pdf-text-excel-files/graphical-enhancement-support.pdf>

Please return your modified manuscript within 60 days; if you cannot complete the modification within this time period, please contact me. If you decide that you do not want to modify the manuscript and wish to submit it to another journal, please notify me of your decision immediately so that the manuscript can be formally withdrawn.

To submit the modified manuscript, log onto the eJP submission site at <https://msystems.msubmit.net/cgi-bin/main.plex>. If you cannot remember your password, click the "Can't remember your password?" link and follow the instructions on the screen. Go to Author Tasks and click the appropriate manuscript title to begin the resubmission process. The information you entered when you first submitted the paper will be displayed. Please update the information as necessary. Provide (1) point-by-point responses to the issues raised by the reviewers as file type "Response to Reviewers," not in your cover letter, and (2) a PDF file that indicates the changes from the original submission (by highlighting or underlining the changes) as file type "Marked Up Manuscript - For Review Only."

To submit your modified manuscript, log onto the eJP submission site at <https://msystems.msubmit.net/cgi-bin/main.plex>. If you cannot remember your password, click the "Can't remember your password?" link and follow the instructions on the screen. Go to Author Tasks and click the appropriate manuscript title to begin the resubmission process (ONLY the corresponding author will have access to the full record for resubmission). The information that you entered when you first submitted the paper will be displayed. Please update the information as necessary and do the following:

- 1) Provide point-by-point responses to the issues raised by the reviewers in a file designated as "Response to Reviewers" (NOT the cover letter).
- 2) Upload ALL of your source files (not PDF and not just the files requiring modification) and make sure that all elements meet the technical requirements for production.
- 3) Do not provide a highlighted or tracked-changes copy of the paper in the main manuscript upload. This should be a clean copy instead. You may provide the compare copy separately by uploading it as a "Marked-Up Manuscript" file.
- 4) Make sure that the figure legends are included in the main manuscript file (not uploaded separately).

If you would like to submit an image for consideration as the Featured Image for an issue, please contact mSystems staff.

Sincerely,

Jack Gilbert
Editor, mSystems

Journals Department
Reviewer comments:

Reviewer #1 (Comments for the Author):

Zund and colleagues present a very well-written review about the current landscape of methodology pertinent to the characterization of phages. This mini-review focuses on cutting edge approaches and will enlighten and entertain readers. I only have a few minor comments for the authors to consider. These comments largely focus on helping the authors improve the clarity and impact of their argumentation.

Line 65: "In fact, the conclusions of our experiments can be dramatically altered when we expand the analysis beyond 67 reference database comparisons, thus capturing a larger fraction of the data". This assertion would benefit from elaboration of the rationale underlying this idea, or a citation that directly supports it.

Line 83: I was somewhat anticipating that this section would discuss the merits of various forms of virus/phage detection beyond read mapping. For example, blast and HMMs have been used to characterize viruses that are too divergent for read mapping approaches to be effective. DeRisi's work in this area, for example: <https://doi.org/10.1371/journal.pone.0105067>

Line 140: It isn't totally clear in this section what the authors mean by "unassignable reads". At first glance, it seemed that they are referring to reads that have no clear viral taxonomic assignment, but later in the paragraph it seems to point to reads that may or may not be viral in origin (as opposed to a specific viral taxon). Related to this, it is not clear how CRISPR-spacers clarify viral taxonomy (and thus read assignment); the point about host range is good, but doesn't seem to clearly link to the other ideas in this paragraph (and possibly belongs in the section starting line 184), possibly because of ambiguity about what "unassignable reads means". It would help to add some specificity here.

Line 144: "One low-cost approach for identifying phages within the unassignable bulk reads is to pool samples prior to virion enrichment and sequencing, which provides higher quality reads for low abundance viruses." While I think I follow, I'm a little confused about how this strategy improves read quality and how that quality then impacts taxonomic assignment of these low abundance viruses. Maybe the authors mean that it improves the coverage of genomes from these taxa in the resulting sequence data? Perhaps the authors could improve the clarity of their argument by being more explicit here, with the goal of advancing motivation for this approach.

Line 184: There appear to be some very powerful computational and analytical solutions to this problem as well that might be worth noting. For example: <https://doi.org/10.1038/s41598-021-81063-4>

Line 202: The section on phage metabolomics is fascinating. However, when reading the manuscript, this section appears to arrive out of left field. The prior sections were motivated by a brief discussion about the importance of characterizing phage identity and host range. Maybe a brief introduction into this section about the importance of characterizing phage functioning, which can be revealed through phage metagenomics, would help this passage fit into the narrative arch of the manuscript.

Line 265: I appreciate the call to the community regarding what can be done to improve the study of phages, especially uncharacterized phages. It seems that it would be worth underscoring a few of the ideas discussed in the preceding sections, notably that researchers should actively deposit novel phage genomes that are resolved from metagenomic investigations into the alternative databases mentioned previously.

Reviewer #2 (Comments for the Author):

Very nice review on the importance of phage "dark matter" and state-of-the-art methods for detecting and identifying phage in complex microbial communities (a.k.a. microbiomes). In addition to NGS-based tools (e.g. metagenomics), the authors expand into novel/emerging technologies including proximity ligation and phage metabolomics. Constructive criticism:

1) proximity ligation seems to be particularly well-poised to revolutionize our understanding of host-phage biology. In one study (<https://doi.org/10.1101/2021.06.14.448389>) the authors identified 100's of host-phage "linkages", thereby more than doubling the total number of host assignments currently known. Proximity ligation seems to offer an important and potentially transformative approach to studying host-phage biology. While the authors do mention proximity ligation, it seemed to be dismissed rather quickly.

2) phage metabolomics is indeed a promising approach towards understanding host-phage biology. However, this approach seems to be somewhat non-sequitur with the rest of the content. Namely, the content of this review is primarily focused on methods to detect and identify phage in microbiomes. Whereas the phage metabolomics is more of a functional measurement. Furthermore, it's not clear how one would systematically perform a phage metabolomics experiment. Would you first need to isolate and purify a phage - then measure the metabolomic profile of a community with and without the addition of the phage? Or is phage metabolomics only useful for understanding interactions between a single host (strain) and a single phage? The authors should make it clearer how phage metabolomics relates to the running theme of this review: namely phage detection and identification. Or if phage metabolomics doesn't apply to this, make it clear that this is a different application space.

Response to Reviewer for "What lies beneath? Taking the plunge into the murky waters of phage biology" (control no. mSystems00807-22 R0)

Dear Editor,

Thank you for the constructive comments from the reviewers. We have gone through the comments and made changes as annotated below.

Reviewer #1 (Comments for the Author):

Zund and colleagues present a very well-written review about the current landscape of methodology pertinent to the characterization of phages. This mini-review focuses on cutting edge approaches and will enlighten and entertain readers. I only have a few minor comments for the authors to consider. These comments largely focus on helping the authors improve the clarity and impact of their argumentation.

Re: We thank the reviewers for their time to review this minireview. We are grateful for the positive comments and constructive suggestions, which have improved our manuscript substantially. Please see below for responses to the individual points that were raised.

Line 65: "In fact, the conclusions of our experiments can be dramatically altered when we expand the analysis beyond 67 reference database comparisons, thus capturing a larger fraction of the data". This assertion would benefit from elaboration of the rationale underlying this idea, or a citation that directly supports it.

Re: Thank you for your comment. We agree that the rationale underlying these ideas needed to be carved out more precisely.

Action taken: We elaborated the rationale in more detail.

Line 83: I was somewhat anticipating that this section would discuss the merits of various forms of virus/phage detection beyond read mapping. For example, blast and HMMs have been used to characterize viruses that are too divergent for read mapping approaches to be effective. DeRisi's work in this area, for example: <https://doi.org/10.1371/journal.pone.0105067>

Re: We thank the reviewer for the feedback and for pointing out that phage identification by profile models like HMM was missing. We believe that the section header "The uncultivable phage fraction is identified mainly through bulk and virome sequencing" misled the reader into anticipating a different section content.

Action taken: We changed the section's header to better represent this central message. Moreover, we incorporated protein alignments and profile methods into the section "The uncultivable phage fraction is identified mainly through bulk and virome sequencing" to highlight its utility for detecting novel phages.

Line 140: It isn't totally clear in this section what the authors mean by "unassignable reads". At first glance, it seemed that they are referring to reads that have no clear viral taxonomic assignment, but later in the paragraph it seems to point to reads that may or may not be viral in origin (as opposed to a specific viral taxon). Related to this, it is not clear how CRISPR-spacers clarify viral taxonomy (and thus read assignment); the point about host range is good but doesn't seem to clearly link to the other ideas in this paragraph (and possibly belongs in the section starting line 184), possibly because of ambiguity about what "unassignable reads means". It would help to add some specificity here.

Re: We agree and thank the reviewer for spotting the unclear meaning of unassignable reads.

Action taken: We defined earlier in the text what we mean by unassignable reads.

Improved definition now appearing in the revised manuscript:

"Even now, with all of the advantages offered by high throughput sequencing, up to 90% of viral metagenomic sequence reads are unassignable, meaning they lack matches to nucleotide or protein databases (5, 6), inspiring terms such as "viral dark matter".

Moreover, we clarify the main take-home message of this paragraph, which included removing the idea that unassignable reads are more likely to originate from phage in virome than in bulk data and moving the CRISPR-spacer part to the subsection: "Targeted experimental approaches to expand knowledge about phage-host pair. ", as suggested by the reviewer.

Improved section:

"Reducing the unassembled fraction of bulk metagenome reads by the targeted use of virome data

De-novo assembly of bulk data does not guarantee the inclusion of all reads to bacterial and viral genomes. Reads from low abundance phages will likely not have adequate coverage for assembly and will be missed during phage identification if sequencing depth is not drastically increased. Considering the abundance of phages in our environment, many unassembled reads likely originate from low abundance virions (47). One low-cost approach for identifying phages within the unassembled bulk reads is to pool samples before virion enrichment and sequencing, which enables improved coverage for low abundance viruses. Mapping bulk data to the viral contigs generated from the pooled virome sample might identify low abundance viruses in individual bulk data allowing us to identify otherwise missed phages."

Line 144: "One low-cost approach for identifying phages within the unassignable bulk reads is to pool samples prior to virion enrichment and sequencing, which provides higher quality reads for low abundance viruses." While I think I follow, I'm a little confused about how this strategy improves read quality and how that quality then impacts the taxonomic assignment of these low abundant viruses. Maybe the authors mean that it improves the coverage of genomes from

these taxa in the resulting sequence data? Perhaps the authors could improve the clarity of their argument by being more explicit here, with the goal of advancing motivation for this approach.

Re: Thank you for pointing out the ambiguity in our language. In this section, we intend to highlight ways to increase the feasibility of identifying phages that are otherwise unassignable within the bulk reads due to low coverage. We suggested generating one pooled virion sample per study to improve the assembly of low abundant virions. Back-mapping reads from bulk samples to contigs generated from this virome sample might allow identifying low abundant phages with individual bulk samples otherwise missed.

Action taken: We clarified our argument in the revised version of the minireview. Revised section:

“De-novo assembly of bulk data does not guarantee the inclusion of all reads to bacterial and viral genomes. Reads from low abundance phages will likely not have adequate coverage for assembly and will be missed during phage identification if sequencing depth is not drastically increased. Considering the abundance of phages in our environment, many unassembled reads likely originate from low abundance virions (47). One low-cost approach for identifying phages within the unassembled bulk reads is to pool samples before virion enrichment and sequencing, which enables improved coverage for low abundance viruses. Mapping bulk data to the viral contigs generated from the pooled virome sample might identify low abundance viruses in individual bulk data allowing us to identify otherwise missed phages.”

Line 184: There appear to be some very powerful computational and analytical solutions to this problem as well that might be worth noting. For example: <https://doi.org/10.1038/s41598-021-81063-4>

Re: We agree with the reviewer that in recent years many powerful new computational tools have been developed to predict phage-host pairs. Rather than discussing the intricate details of different bioinformatic solutions, we point the reader to two extensive and excellent reviews, with one just published recently (Coclet C, Roux S., Curr Opin Virol 2021 and Edwards RA, McNair K, Faust K, Raes J, Dutilh BE., FEMS Microbiol Rev 2015.).

Action taken: As suggested by the reviewer before, we moved the CRISPR-spacer example into this section. Moreover, we used the suggested study as an example to predict phage-host pairs at the protein level.

Line 202: The section on phage metabolomics is fascinating. However, when reading the manuscript, this section appears to arrive out of left field. The prior sections were motivated by a brief discussion about the importance of characterizing phage identity and host range. Maybe a brief introduction into this section about the importance of characterizing phage functioning, which can be revealed through phage metagenomics, would help this passage fit into the narrative arch of the manuscript.

Re: We agree that improvement is needed to better incorporate the phage metabolomics section into the minireview.

Action taken: We adapted the phage metabolomics section to better emphasize the crucial role of phage metabolomics in understanding the phage-bacterial interaction. Phage metabolomics is one route for moving beyond identification into function.

Improved phage metabolomics introduction:

“Reliant upon the biological mechanics of their hosts, phages cannot autonomously perform many processes considered fundamental to life, including metabolism. Phages must therefore rely entirely on their hosts for the production of all necessary biomolecules. Although much can be learned through functional characterization of a phage genome, a more comprehensive understanding of the metabolic consequences of a phage infection can be achieved through metabolomics experiments conducted over the course of a phage infection. In some instances, novel phages may even be revealed through an examination of “phage-induced” metabolic phenotypes that arise in uncharacterized multicomponent systems. We further argue that, just as the identification of a novel phage is incomplete without a specific link to a host, the characterization of a phage is incomplete without understanding its impact on host metabolism. We, therefore, broadly define phage metabolomics as “the study of phage-mediated metabolic changes in bacteria”, a definition first espoused by others (e.g., De Smet 2016 (69)).“

Line 265: I appreciate the call to the community regarding what can be done to improve the study of phages, especially uncharacterized phages. It seems that it would be worth underscoring a few of the ideas discussed in the preceding sections, notably that researchers should actively deposit novel phage genomes that are resolved from metagenomic investigations into the alternative databases mentioned previously.

Re: We are happy to hear that the reviewer likes our shout-out to the community, and we appreciate the suggestions for improvement.

Action taken: In the discussion, we emphasize that the community should make a combined effort to deposit uncultivated phage genomes in alternative databases. Moreover, we also highlight the underlying problem that many alternative databases are not maintained, thus only presenting a snapshot of a specific time. The maintenance of databases for uncultivated virus sequences will be crucial to driving the field into the next century of virus research.

Reviewer #2 (Comments for the Author):

Very nice review on the importance of phage "dark matter" and state-of-the-art methods for detecting and identifying phage in complex microbial communities (a.k.a. microbiomes). In addition to NGS-based tools (e.g. metagenomics), the authors expand into novel/emerging technologies, including proximity ligation and phage metabolomics. Constructive criticism:

1) proximity ligation seems to be particularly well-poised to revolutionize our understanding of host-phage biology. In one study (<https://doi.org/10.1101/2021.06.14.448389>) the authors identified 100's of host-phage "linkages", thereby more than doubling the total number of host

assignments currently known. Proximity ligation seems to offer an important and potentially transformative approach to studying host-phage biology. While the authors do mention proximity ligation, it seemed to be dismissed rather quickly.

Re: We agree with the reviewer that proximity ligation is a promising approach to improve the phage-host pair assignment.

Action taken: We expanded the proximity ligation section and highlighted how this technology will advance the field.

2) phage metabolomics is indeed a promising approach towards understanding host-phage biology. However, this approach seems to be somewhat non-sequitur with the rest of the content. Namely, the content of this review is primarily focused on methods to detect and identify phage in microbiomes. Whereas the phage metabolomics is more of a functional measurement. Furthermore, it's not clear how one would systematically perform a phage metabolomics experiment. Would you first need to isolate and purify a phage - then measure the metabolomic profile of a community with and without the addition of the phage? Or is phage metabolomics only useful for understanding interactions between a single host (strain) and a single phage? The authors should make it clearer how phage metabolomics relates to the running theme of this review: namely phage detection and identification. Or if phage metabolomics doesn't apply to this, make it clear that this is a different application space.

Re: We thank the reviewer for pointing out that we must clearly outline how phage metabolomics relates to phage genomics.

Action taken: We adapted the introduction to the phage metabolomics section to better emphasize the crucial role of phage metabolomics in understanding the phage-bacterial interaction. Phage metabolomics is one route for moving beyond identification into function.

Improved introduction:

“Reliant upon the biological mechanics of their hosts, phages cannot autonomously perform many processes considered fundamental to life, including metabolism. Phages must therefore rely entirely on their hosts for the production of all necessary biomolecules. Although much can be learned through the functional characterization of a phage genome, a more comprehensive understanding of the metabolic consequences of a phage infection can be achieved through metabolomics experiments conducted over the course of a phage infection. In some instances, novel phages may even be revealed through an examination of “phage-induced” metabolic phenotypes that arise in uncharacterized multicomponent systems. We further argue that, just as the identification of a novel phage is incomplete without a specific link to a host, the characterization of a phage is incomplete without understanding its impact on host metabolism. We, therefore, broadly define phage metabolomics as “the study of phage-mediated metabolic changes in bacteria”, a definition first espoused by others (e.g., De Smet 2016 (69)). “

November 2, 2022

Prof. Katrine Whiteson
University of California, Irvine
Department of Molecular Biology and Biochemistry
School of Biological Sciences
3236 McGaugh Hall
Irvine, CA 92697-3900

Re: mSystems00807-22R1 (What lies beneath? Taking the plunge into the murky waters of phage biology)

Dear Prof. Whiteson:

Your manuscript has been accepted, and I am forwarding it to the ASM Journals Department for publication. For your reference, ASM Journals' address is given below. Before it can be scheduled for publication, your manuscript will be checked by the mSystems production staff to make sure that all elements meet the technical requirements for publication. They will contact you if anything needs to be revised before copyediting and production can begin. Otherwise, you will be notified when your proofs are ready to be viewed.

Publication Fees:

If you would like to submit a potential Featured Image, please email a file and a short legend to mSystems@asmusa.org. Please note that we can only consider images that (i) the authors created or own and (ii) have not been previously published. By submitting, you agree that the image can be used under the same terms as the published article. File requirements: square dimensions (4" x 4"), 300 dpi resolution, RGB colorspace, TIF file format.

We recognize that the video files can become quite large, and so to avoid quality loss ASM suggests sending the video file via <https://www.wetransfer.com/>. When you have a final version of the video and the still ready to share, please send it to mSystems staff at mSystems@asmusa.org.

Sincerely,

Jack Gilbert
Editor, mSystems

Journals Department
E-mail: mSystems@asmusa.org